# The Production of Recombinant African Swine Fever Virus Lv17/WB/Rie1 Strains and Their In Vitro and In Vivo Characterizations

**DOI:** 10.3390/vaccines11121860

**Published:** 2023-12-17

**Authors:** Stefano Petrini, Cecilia Righi, István Mészáros, Federica D’Errico, Vivien Tamás, Michela Pela, Ferenc Olasz, Carmina Gallardo, Jovita Fernandez-Pinero, Eszter Göltl, Tibor Magyar, Francesco Feliziani, Zoltán Zádori

**Affiliations:** 1National Reference Centre for Pestiviruses and Asfivirus, Istituto Zooprofilattico Sperimentale Umbria-Marche “Togo Rosati”, Via Gaetano Salvemini, 1, 06126 Perugia, Italy; s.petrini@izsum.it (S.P.); c.righi@izsum.it (C.R.); f.derrico@izsum.it (F.D.); m.pela@izsum.it (M.P.); 2HUN-REN Veterinary Medical Research Institute (VMRI), Hungária krt. 21, 1143 Budapest, Hungary; meszaros.istvan@vmri.hun-ren.hu (I.M.); tamas.vivien@vmri.hun-ren.hu (V.T.); olasz.ferenc@vmri.hun-ren.hu (F.O.); goltl.eszter@vmri.hun-ren.hu (E.G.); magyar.tibor@vmri.hun-ren.hu (T.M.); 3European Union Reference Laboratory for ASF (EURL-ASF), Centro de Investigación en Sanidad Animal (CISA-INIA, CSIC), Valdeolmos, 28130 Madrid, Spain; gallardo@inia.csic.es (C.G.);

**Keywords:** African swine fever, ASFV, live-attenuated vaccine, illegitimate recombination, homologous recombination, pigs

## Abstract

Lv17/WB/Rie1-Δ24 was produced via illegitimate recombination mediated by low-dilution serial passage in the Cos7 cell line and isolated on PAM cell culture. The virus contains a huge ~26.4 Kb deletion in the left end of its genome. Lv17/WB/Rie1-ΔCD-ΔGL was generated via homologous recombination, crossing two ASFV strains (Lv17/WB/Rie1-ΔCD and Lv17/WB/Rie1-ΔGL containing eGFP and mCherry markers) during PAM co-infection. The presence of unique parental markers in the Lv17/WB/Rie1-ΔCD-ΔGL genome indicates at least two recombination events during the crossing, suggesting that homologous recombination is a relatively frequent event in the ASFV genome during replication in PAM. Pigs infected with Lv17/WB/Rie1-Δ24 and Lv17/WB/Rie1/ΔCD-ΔGL strains have shown mild clinical signs despite that ASFV could not be detected in their sera until a challenge infection with the Armenia/07 ASFV strain. The two viruses were not able to induce protective immunity in pigs against a virulent Armenia/07 challenge.

## 1. Introduction

African swine fever (ASF) is a devastating and economically significant disease of *Sus scrofa* caused by a DNA arbovirus; it affects both domesticated pigs and wild boars. The ASF virus (ASFV) is a unique member of the family *Asfarviridae* and has a linear dsDNA genome of 170–194 kbp [1].

To date, there is no safe and effective vaccine against ASFV on the market despite that it can reach 100% mortality, so the best way to control the virus is through effective biosecurity measures, rapid detection, and the culling of infected animals. The total economic loss caused by ASFV is difficult to estimate. The endemic ASF in Africa causes significant economic hardships for local pig farmers and affects hundreds of farms. The economic cost of ASF between 2014 and 2018 was estimated at USD 1,500,000 in Benin and USD 900,000 in Nigeria in 2001 [2].

In 2022, a new outbreak of ASF was reported in industrial pig farms in Russia, with nearly 20,000 pigs culled, and the Russian pig industry suffered a loss of USD8.5 million due to the outbreak [3]. In Asia, more than 200 million pigs have been slaughtered due to control measures [4]. In Vietnam alone, the economic damage is estimated to be between USD 880 million and USD 4.4 billion [2,5]. Recent outbreaks of ASFV in various regions of Italy have caused great concern, leading to a ban on the export of pork products [6].

The ASFV Lv17/WB/Rie1 was discovered in Latvia in 2017; it was isolated from the serum of a hunted wild boar. Lv17/WB/Rie1 is a naturally attenuated isolate and grows very efficiently in macrophage cell cultures. Sequencing the whole genome of the virus identified a single nucleotide deletion within the EP402R gene [7]. The mutation rendered the encoded CD2-like protein non-functional, and the isolate lost its ability to induce hemadsorption (non-HAD isolate) [8]. Lv17/WB/Rie1 in pigs causes mild symptoms (swelling of joints and cyanosis in ears) or remains subclinical. Experimental infection with the virus caused a short viremia and induced a high anti-ASFV antibody titer, but infected pigs shed the virus and transmitted it to contact pigs [7]. The oral immunization of wild boar with the virus resulted in high protection (92%) against virulent ASFV challenge [9]. These experiments suggest that, after further targeted modifications, it could serve as a basis for an effective modified live virus (MLV) vaccine. Indeed, the deletion of the MGF 110–11L gene reduced the side effects of the virus, but not to the desired level [10].

The ASFV 9GL gene (or B119L) encodes a late, nonstructural 14-kDa protein. 9GL is highly conserved at protein and DNA levels in any virus isolate. The 9GL-encoded protein (p9GL) is involved in normal virion maturation, and viral growth in vitro, and can affect virulence in vivo [11]. The 9GL protein (p9GL) belongs to the Erv1p/Alrp family and has FAD-linked sulfhydryl oxidase activity [12]. The deletion of the 9GL gene resulted in varying degrees of attenuation depending on the ASFV strain [11,13,14].

The 8-DR (or EP402R) gene encodes a 402-amino-acid-long, late-transcripted protein (pEP402R or CD2v/CD2-like protein) that is homologous to the CD2 protein of T cells. CD2v is a type I transmembrane protein and is involved in the HAD of red blood cells to infected macrophages [8]. Experiments involving deletion of the EP402R gene suggest that the protein may play a role in the immunosuppression of ASFV [15].

pEP153R is a 153-amino-acid-long early- and late-transcribed transmembrane protein encoded by the EP153 (8CR) open reading frame. The protein consists of a C-type lectin domain and is involved in stabilizing the interaction between the viral CD2v protein and its cell receptor. Previous results suggested that EP153R is also involved in HAD, but recent data seem to contradict this [16,17]. pEP153R also inhibits the host cell’s MHC-I membrane expression [18] and influences apoptosis by inhibiting caspase-3 activation [19].

The targeted modification of ASFV is facilitated by homologous recombination. Cas9-induced double-strand breaks induce a relatively high-frequency recombination between ASFV and transfer plasmids [20], and some indirect evidence suggests that homologous recombination contributes to the evolution of ASFV [21]. However, to the best of our knowledge, experimental evidence of homologous recombination between ASFV genomes has not yet been published.

The development of MLV is not only hampered by the laborious task of creating the right virus but also by the lack of cell lines that support the unrestricted growth of ASFV. The adaptation of field strains or vaccine candidates almost always involves significant genetic changes including point mutations and deletions located mostly in the variable regions of the virus and frequently the alteration of the immunogenic properties of the parental strains [22]. The size of deletions can reach tens of kilobases after serial passages [23].

For these reasons, in the current study, we have conducted different experiments with the following objectives: (i) examine the stability of Lv17/WB/Rie1 in the Cos7 cell line and follow how serial passages affect the stability of the viral genome; (ii) determine whether it is possible to generate and isolate recombinant viruses in porcine alveolar macrophage (PAM) cells by coinfection; (iii) evaluate the safety and efficacy of two of the vaccine candidates generated by these methods in in vivo experiments. 

## 2. Materials and Methods

### 2.1. Cell Culture

The PAMs were prepared following guidelines set out in the WOAH manuals [24]. They were frozen in RPMI-1640 medium (Thermo Scientific, Waltham, MA, USA), supplemented with 30% fetal bovine serum (Sigma-Aldrich, Saint Louis, MO, USA) and 10% DMSO, and stored at −72 °C. For infection, cells were cultured in RPMI-1640 medium containing 1X antibiotic/antimycotic solution (Thermo Scientific, Waltham, MA, USA) with 2 mM L-glutamine (Sigma-Aldrich, Saint Louis, MO, USA) and 10% (*v*/*v*) heat-inactivated fetal bovine serum (FBS) (Thermo Scientific, Waltham, MA, USA). Prior to infection, PAMs were incubated at 37 °C in 5% CO_2_ for 24 h.

Cos7 cell culture was originally obtained from the American Type Culture Collection (ATCC CRL-1651), and it was grown in Dulbecco’s modified Eagle Medium (DMEM) (Capricorn, Ebsdorfergrund, Germany) supplemented with 2 mM L-glutamine, 100U of gentamicin per mL (Sigma-Aldrich, Saint Louis, MO, USA), 1% Na Pyruvate (Thermo Scientific, Waltham, MA, USA), and 1% non-essential amino acids (Thermo Scientific, Waltham, MA, USA). Cells were cultured at 37 °C in medium supplemented with 10% FBS.

### 2.2. Production of Lv17/WB/Rie1-Δ24 Virus

The ASF Lv17/WB/Rie1-Δ24 (GeneBank Accession Number: OR806651) was created by serial passage. The first 5 passages were performed in Cos7 cell culture (ATCC CRL-1651) and were carried out at the European Union Reference Laboratory (EURL) for ASF, Centro de Investigación en Sanidad Animal (CISA/INIA-CSIC, Madrid, Spain). In short, Cos7 cells were infected with ASFV Lv17/WB/Rie1 (GeneBank Accession Number: OR863253) at an MOI of 3 PFU/cell in FBS-free media. Following 120 min incubation at 37 °C for viral adsorption, the medium was supplemented with 2% FBS, and cells were incubated at 37 °C in for 7 days. After incubation, cells were harvested, subjected to three freeze–thaw cycles, and clarified by centrifugation at 2000× *g* for 10 min at 4 °C. The resulting supernatants were filtered through a 0.22 μm filter to eliminate cellular impurities. Subsequently, the virus in the cultured fluid was sub-passaged 5 times, following a similar procedure. Viral replication was identified by real time PCR (WOAH 2021) from 200 μL of the cultured supernatant obtained in each of the passages. The virus was analyzed after 5 passages and named Lv17_Cos5. Further passages were carried out in the VMRI, as follows.

Cos7 cells were maintained in DMEM medium with a high glucose concentration and supplemented with 10% (*v*/*v*) fetal bovine serum (Sigma-Aldrich, Saint Louis, MO, USA) and 1X antibiotic/antimycotic solution (Thermo Scientific, Waltham, MA, USA). The cells were incubated at 37 °C in 5% CO_2_. For the next three passages, 200 uL of supernatant (10× dilution) was transferred every 7th day to a new well of freshly plated Cos7 cells. After 8 passages (Lv17_cos8), the next 3 ones were performed in PAMs (Lv17_cos8_pam3). Every 3 days, 200 uL of supernatant was transferred to fresh PAMs. After the 11th passage, the next two ones were also performed in PAMs by infecting the cells with a tenfold dilution series of the supernatants, and only the highest dilution, where complete cell death was observed, was further passaged. The 14th (last) passage (Lv17/WB/Rie1-Δ24) was performed in a 75 cm^2^ flask with 0.2 MOI, and the supernatant was harvested at 72 hpi.

### 2.3. Production of Lv17/WB/Rie1-ΔCD-ΔGL Virus

The ASFV Lv17/WB/Rie1-ΔCD-ΔGL (GeneBank Accession Number: OR806652) was created through homologous recombination from the mCherry marker-containing B119L gene-deleted Lv17/WB/Rie1-ΔGL strain and the eGFP gene-containing EP402R and EP153R-deleted Lv17/WB/Rie1-ΔCD mutant virus.

Lv17/WB/Rie1-ΔGL and Lv17/WB/Rie1-ΔCD stocks were prepared according to Tamás et al. (2023) [10]. PAMs were cultured as described above and infected with 3-3 MOI from Lv17/WB/Rie1-ΔGL and Lv17/WB/Rie1-ΔCD stocks. After 1 h incubation, supernatant was removed and replaced with a fresh PAM culture medium. The supernatant was collected 3 days after infection. The recombinant viruses were serially purified by pipetting red-green cells under a fluorescent microscope, freeze–thawing and plating them after dilution. When a virus isolate was considered homogeneous, two more purification steps were carried out with high dilution and the “final stocks” were established.

### 2.4. Determination of Growth Characteristics

To determine the specific infectivity (SI) of the different viral strains, PAM cells were infected with 0.01 MOI and sampled every 12 h for 108 h. SI was calculated as the ratio of the total viral genome copy to the number of infectious viruses, measured in FFUs (SI = genome copy/FFU). Viral titers and ASFV genome copies were determined as described by Tamás et al. (2023) [10]. Briefly: p72-specific PCR was performed for quantification. The qPCR (25 μL) was performed with 1 µL of template DNA from the diluted supernatants and 1 µL forward (F2: 5′-TACGTTGCGTCCGTGATAGG-3′) and 1 µL reverse (R2: 5′-AGTTCGGATGTCACAACGCT-3′) primers at a concentration of 1 µM, 12.5 µL of DreamTaq PCR Master Mix (2×; Thermo Fisher Scientific, Waltham, MA, USA), and 20× EvaGreenTM Dye (Biotium, Fremont, CA, USA). The diluted supernatants of infected cells were used directly in the reaction after heat treatment (72 °C, 20 min). The thermal reaction was initiated with a 5 min pre-denaturation step at 95 °C, followed by 35 cycles containing a denaturation step at 95 °C for 30 s, followed by annealing at 62 °C for 30 s, elongation at 72 °C for 35 s, and a post-elongation step at 72 °C for 5 min. In each case, the specificity of the reaction was verified using melting curve analysis. Viral copy numbers were calculated using a standard curve of 10-fold dilution of the purified amplicon as a template.

### 2.5. Immunfluorescent Staining

To fix PAM cells, 3% formaldehyde (VWR International, Radnor, PA, USA) diluted in 1× PBS (Capricorn Scientific GmbH, Ebsdorfergrund, Germany) was used. Mouse Anti-p72 monoclonal antibody (Ingenasa, Madrid, Spain) at 40× dilution and Highly Cross-Adsorbed CF594 labelled Goat Anti-Mouse IgG (H + L) secondary antibody (Biotium, Fremont, CA, USA) at 1000× dilution were used for immunofluorescent staining. Positive cells were visualized under a an Axio Observer D1 inverted fluorescence microscope (Carl Zeiss Ag. Oberkochen, Germany). Infectious virion concentration was calculated from the dilution ratio and the number of infected cells per well and was expressed in Fluorescence Focus Unit (FFU)/mL.

### 2.6. Sequencing

Full genome sequencing of all strains and isolates was performed according to Olasz et al. (2019) [25]. Sequences were assembled with Geneious Prime 2019.2.3 using the Bowtie2 mapping method with normal sensitivity for the analysis. The reference strain was the original Lv17/WB/Rie1 strain (patented in Spain under reference PCT/2018/000069). Deletion breakpoints were confirmed using split read alignments and de novo assembly.

### 2.7. Animal Experiments

Thirteen three-month-old crossbreed pigs, confirmed healthy and ASFV-free, were kept in the BSL3 animal facilities of the Istituto Zooprofilattico Sperimentale Umbria-Marche, Perugia, Italy. They received a diet designed for fattening pigs twice daily and had unrestricted access to water. All care and experimental procedures followed the European regulations safeguarding animals used for scientific research. The experiments were carried out with approval of the Italian Ministry of Health (No. 424/2020-PR).

The determination of the “Human End Point” (HEP), indicating the severity of injury, was evaluated using welfare indicators specified in a document sanctioned by the competent national authorities, in accordance with the implementation guidelines of EU Directive 2010/63. Additionally, considerations were given to the directives outlined in the Working Document on a Severity Scoring System formulated by a panel of experts. Specifically, the HEP was determined by exceeding a singular criterion (such as absence of food or water intake for 24 h, or hypothermia for 24 h, or immobility for 24 h) or by achieving an overall score exceeding 17.

Before the start of the experiments, the pigs were acclimatized for seven days, and the animals were divided into three groups. The first two groups consisted of five animals in each group, while the third group consisted of three animals. The first two groups were injected with the vaccine candidate Lv17/WB/Rie1-Δ24 (Lv17/WB/Rie1-Δ24 group) and Lv17/WB/Rie1-ΔCD-ΔGL (Lv17/WB/Rie1-ΔCD-ΔGL group). Two administrations of each group were carried out 21 days apart. For both vaccine candidates, a dose of 10^2^ FFU/mL was used in the first vaccination, while a dose of 10^4^ FFU/mL was used in the second immunization. The third group (control group) represented the unvaccinated controls (Table 1). The vaccine candidates received an injection intramuscularly into the neck muscle (right side).

Forty-nine days post vaccination (DPV), all animals were subjected to a challenge infection with the Armenia/07 strain of ASFV (99.7% homology to Lv17/WB/Rie1). Each pig received 10^2^ HAD_50_/2 mL administered via the intramuscular route into the neck muscle (right side).

The pigs were observed for 7 days after the challenge infection, and clinical signs were recorded daily using a clinical score table developed earlier [10,26].

Serum and blood samples were collected at different times: 0, 6, 10, 21, 31, 35, 41 and 49 days post-vaccination (DPV) and 0, 4, and 7 days post challenge (DPC). Antibodies to ASFV have been detected in sera by ELISA tests, while viremia to ASFV has been detected in blood samples by real-time PCR. Surviving pigs were euthanized and necropsied at the end of the experiments.

### 2.8. Collection of Blood Samples

Approximately 7 mL of blood samples were drawn from the jugular vein of each pig using single-use needles and Vacutainers (Kima, Padova, Italy). Samples were centrifuged at 850× *g* for 30 min at 4 °C to separate the sera for serological analysis. Subsequently, all samples were stored at −20 °C for further serological analysis.

### 2.9. ELISA Test

An ELISA kit (ID Screen African Swine Fever Competition, Grables, France) was used to detect anti-ASFV antibodies. Interpretation of the results followed the manufacturer instructions. Specifically, the competition percentage (S/N) for each sample was calculated based on the following criteria: (i) positive: S/N % ≤ 40%; (ii) doubtful: 40% < S/N % < 50%; (iii) negative: S/N % ≥ 50%. An automated plate reader (Infinite F50, Tecan AG, Männedorf, Switzerland) was used to read the microplates, and the data were analyzed using Magellan software version 7.1 (Tecan AG, Männedorf, Switzerland).

### 2.10. Real-Time PCR

The High Pure PCR Template Preparation Kit was employed to extract viral DNA from the blood samples following the manufacturer’s guidelines. To detect the presence of ASFV, real-time PCR was performed in accordance with the procedures outlined in the Manual of Diagnostic Tests and Vaccines for Terrestrial Animals [24].

### 2.11. Statistical Analysis

For the statistical analysis of specific infectivity, the Friedman test was used. A threshold of *p* < 0.05 was set to determine statistical significance. The R software package (version 4.2.2) was used for all calculations.

## 3. Results

### 3.1. Generation of Lv17/WB/Rie1-Δ24

To study the genomic stability of Lv17/WB/Rie1 in a permissive established cell line, we serially passaged the virus in Cos7 cells. The successful replication of Lv17/WB/Rie1 in these cells was demonstrated during the first five passages. The Ct values of ASFV DNAs were 20.36, 19.58, 16.65, 16.97, and 16.67 at passages 1, 2, 3, 4, and 5, respectively. To further monitor genetic changes in Cos7, three additional passages were performed. Aliquots were taken at the fifth (Lv17_cos5) and eighth (Lv17_cos8) Cos7 passages, and Lv17_cos8 was reintroduced into PAMs and then passaged three more times to generate Lv17_cos8_pam3 viral stock. Viruses from the three stocks were deep-sequenced on the Illumina platform. Around 3, 2.8, and 2.2 million viral reads were gained from the two “Cos7” and the “PAM” stocks, respectively, and used to assemble and analyze the virus sequences.

For all three viruses, relatively homogeneous coverage was observed (average coverage was 1912, 2274, and 1719 for Lv17_cos5, Lv17_cos8, and Lv17_cos8_pam3, respectively), with a decrease towards the ends of the genomes, but even in the distal 100 nucleotides, the minimum coverage exceeded 40 in each case (27, 207, and 43, respectively). Sequence analysis of the three stocks revealed that Lv17/WB/Rie1 suffered significant genome rearrangements in the form of large indels in the left variable region (LVR), but only three point mutations could be detected in the passages analyzed (a G insertion at 189,481 (non-coding) in all passages, a C-to-T transition in the M1249L gene at 77,191 in the two Cos7 passages, and a G-to-A transition in the MGF 360 11L gene at 27,611 in the Lv17_cos8_pam3 passage).

In the Lv17_cos5 and Lv17_cos8 stocks, the major viral component of the quasispecies lost around 40 kb (2352–40,568) of its left terminus, encoding at least 52 genes, and was substituted by a viral fragment of nearly similar size (40,447 bp, coding at least 51 genes) translocated from the right terminus of the genome (Table 2). In both passages, a relatively small number of viral reads (907 and 3 reads in Lv17_cos5 and Lv17_cos8, respectively) could still be mapped to the region of the genome between the proximal endpoint of the inverted terminal repeat and the deletion breakpoint (2352 and 40,568, respectively), indicating the presence of other genotypic variants as minor components in the virus mix (Figure 1).

The titer of Lv17_cos8 in PAM was very low <10^4^/mL in the first passage and then gradually increased and exceeded 10^6^/mL in the third passage. In Lv17_cos8_pam3, a much longer deleted version of Lv17/WB/Rie1 became a major variant. The translocation was missing from the right end and contained a much shorter deletion, now ranging from 187 to 26,554 nucleotides (encoding 41 genes) (Table 2). Since 63 reads were still mapped to the deleted region, two additional end dilution/passage cycles were performed to produce a homogeneous population. A high-multiplicity infection with this virus resulted in a seemingly homogeneous stock (Lv17/WB/Rie1-Δ24) (no reads detected between 187 and 26,554) with an even higher titer (4.5 × 10^7^ FFU/mL) than Lv17_cos8_pam3, and it was used to inoculate the animals.

These results clearly indicate that Lv17/WB/Rie1 is genetically instable in Cos7 cells, and the stocks are not homogeneous, with the majority of the viruses losing around 40 kb in only five passages. However, even minor genetic variants that are non-detectable or difficult to detect can persist in low-dilution passages during adaptation and can become the major variant in a few passages under different culturing conditions—in this case, in PAM cells.

### 3.2. Production of Lv17/ΔCD-ΔGL

Since indirect data suggest that homologous recombination between ASFVs can occur between ASFVs, we wondered whether two ASFV strains could be crossed in vitro and whether recombinants could be generated and isolated by such a method.

Therefore, we infected PAM cells with high concentrations of Lv17/WB/Rie1-ΔCD and Lv17/WB/Rie1-ΔGL strains carrying the eGFP and mCherry marker genes. After 3 days, the resulting stock was serially diluted and red-green fluorescent cells were detected under a fluorescence microscope. Approximately 1–2% of the infected cells showed both red-green fluorescence; these cells were isolated and re-plated after isolation. By repeating the isolation and re-plating procedure five times, a homogeneous isolate was obtained in which all infected cells showed red-green fluorescence. After fixation, a viral infection of the red-green cells was confirmed with anti-p72 primary and anti-mouse secondary antibodies. Stock was then prepared from the homogeneous isolate and sequenced. Sequencing confirmed the homogeneity of the virus stock, no reads covering the knocked-out gene sequences (EP402R, EP153R, B119L) were found, which would have indicated the presence of the parental viruses. Compared to the two parental strains, six novel mutations occurred in the left variable region of the genome, including four in the poly G/C regions and two of them in the MGF 300-1L gene, inducing a single amino acid change. More interestingly, the disappearance of the unique sequence markers (A188516G and C188519T) of Lv17/WB/Rie1-ΔGL and the appearance of the unique marker (189481 + G) of Lv17/WB/Rie1-ΔCD in the genome of the recombinant Lv17/WB/Rie1-ΔCD-ΔGL (Figure 2) suggest that, during its formation, at least two crossovers occurred: one between the endpoints of the two fluorescent markers (75384-96072) and one between nucleotides 114,799 and 188,516. All this shows that recombination can and does occur in macrophages between different ASFV strains, and recombinant strains can be produced relatively easily by crossing.

### 3.3. Replication of Lv17/WB/Rie1-ΔCD-ΔGL and Lv17/WB/Rie1-Δ24 Viruses

To examine how the Lv17/WB/Rie1-ΔCD-ΔGL and Lv17/WB/Rie1-Δ24 mutants replicate in vitro, PAMs were infected with Lv17/WB/Rie1-ΔCD-ΔGL and Lv17/WB/Rie1-Δ24 viruses, as well as with Lv17/WB/Rie1 as a control with an MOI of 0.01, and were sampled at various time points (0, 12, 24, 36, 48, 60, 72, 84, 96, and 108 h post infection (hpi)). The collected samples were then titrated in PAMs, and qPCR was used to determine the viral copy numbers in the aliquots. There were significant differences in the growth kinetics of the three viruses. Both mutant viruses showed a slower growth rate than Lv17/WB/Rie1, and their end titers (6.9 × 10^3^ and 2.5 × 10^5^ FFU/mL for Lv17/WB/Rie1-ΔCD-ΔGL and Lv17/WB/Rie1-Δ24, respectively) were significantly lower than that of Lv17/WB/Rie1 (1.8 × 10^6^), at 108 hpi (Figure 3).

Clearly, the genetic modifications of the mutant viruses weakened their ability to replicate in PAM, most probably due to a decrease in their infectivity, as indicated by their specific infectivity (SI) values (Lv17/WB/Rie1 vs. Lv17/WB/Rie1-Δ24: *p* = 0.07817; Lv17/WB/Rie1 vs. Lv17/WB/Rie1-ΔCD-ΔGL: *p* = 0.00395). From the earliest time point at which measurable growth could be detected for all viruses (48 hpi), the SI of the Lv17/WB/Rie1-ΔCD-ΔGL virus was significantly (Lv17/WB/Rie1 vs. Lv17/WB/Rie1-ΔCD-ΔGL: *p* = 0.00395) higher than that of Lv17/WB/Rie1, which varied around 10^3^ genome copies/FFU during the experiment. Although the *p*-value of the SI of the Lv17/WB/Rie1-Δ24 remained slightly above the significance threshold (Lv17/WB/Rie1 vs. Lv17/WB/Rie1-Δ24: *p* = 0.07817), the SI values of Lv17/WB/Rie1-Δ24 almost always exceeded that of the parental Lv17/WB/Rie1. In the late (most intensive) period of the viral growth phase (72–108 hpi), when the measured data were less variable and the standard deviation was smaller, the SI of the worse-growing Lv17/WB/Rie1-ΔCD-ΔGL exceeded that of the wild type by an order of magnitude, while the SI of the Lv17/WB/Rie1-Δ24 mutant remained between the values of the other two viruses, just like its titer (Figure 3).

### 3.4. Evaluation of a Lv17/WB/Rie1-Δ24 and Lv17/WB/Rie1-ΔCD-ΔGL in Pigs

Three pigs in the Lv17/WB/Rie1-Δ24 group showed ecchymosis of the ear pinnae (pig #2; 21 DPV) and agitation (pigs #4–5; 4 DPC). Furthermore, all pigs in the Lv17/WB/Rie1-ΔCD-ΔGL group showed the following clinical symptoms: anorexia (pigs #7–8; 6 DPC); hemorrhagic petechiae of the ear pinnae (pigs #9; 10 DPV); diarrhea (pigs #6, 7, 9, and 10; 2–5 DPV); inappetence (pigs #7–8; 6 DPC). Two pigs in the control group showed inappetence (#11–12; 7 DPC). Four days after challenge, the Lv17/WB/Rie1-Δ24 and Lv17/WB/Rie1-ΔCD-ΔGL groups showed fever (>40.5 °C), while fever (Figure 4) was observed in the control group 6 DPC.

During the vaccination period, the clinical score of the Lv17/WB/Rie1-Δ24 and Lv17/WB/Rie1-ΔCD-ΔGL groups remained below 0.6. Following the challenge infection, the clinical score increased for all groups. In particular, the Lv17/WB/Rie1-Δ24 group showed a score of 1.40, 4 DPC, while the Lv17/WB/Rie1-ΔCD-ΔGL group showed a score of 2.80, 7 DPC. The control group scored 2.00, 6 DPC (Figure 5).

Eight pigs suddenly died after the challenge infection. In particular, two pigs (# 1 and #4) in the Lv17/WB/Rie1-Δ24 group died 4 and 3 DPC. Furthermore, three animals (#6, #7 and #8) from Lv17/WB/Rie1-ΔCD-ΔGL group died 3 DPC, whereas three pigs (#11, #12 and #13) from the control group died from 2 to 4 DPC. None of the remaining pigs had reached the humane end points when they were euthanized at the end of the experiment (7 DPC).

In all observed groups, the necropsy studies revealed lymph nodes (bronchial, submandibular, epigastric, perirenal, mesenteric, superficial inguinal) characterized by enlargement and hemorrhagic lesions as well as spleens showing splenomegaly. In addition, in some animals in the three observed groups, we observed lung lesions referred to acute bronchopneumonia and kidney lesions characterized by superficial point hemorrhages and petechial hemorrhages. In addition, one pig in the Lv17/WB/Rie1-Δ24 group and three pigs in the Lv17/WB/Rie1-ΔCD-ΔGL group showed tonsil lesions characterized by superficial point hemorrhages. In contrast, all animals in the Lv17/WB/Rie1-Δ24 group and the control group showed cardiac lesions characterized by pericardial effusion, superficial point hemorrhages, and petechial hemorrhages. Only one pig in the Lv17/WB/Rie1-ΔCD-ΔGL group showed superficial point hemorrhages at the cardiac level. Lesions at a renal level were only observed in the first two groups and were characterized by diffuse hemorrhages. Finally, some animals in the Lv17/WB/Rie1-Δ24 and Lv17/WB/Rie1-ΔCD-ΔGL groups showed intestinal lesions characterized by hemorrhagic enteritis of the small intestine (Table 3).

One animal in the Lv17/WB/Rie1-Δ24 group evidenced antibodies at 17 DPV. Later, the number of positive pigs increased, and 4 DPC, all animals tested positive. These results were detected until the end of the experiments. Otherwise, in the Lv17/WB/Rie1-ΔCD-ΔGL group, antibodies were detected for the first time 31 DPV. Differently, in controls, the antibodies were not detected during the entire experimental period (Table 4).

The first positive results were detected by real-time PCR in all groups 4 DPC. Positivity was maintained until the end of the study period (7 DPC) (Table 5).

## 4. Discussion

The passage of ASFV in established cell lines usually leads to large deletions [27] in the viral genome, most often in the left and right variable regions (LVR and RVR, respectively). Extensive regions were deleted from the genomes of strain E70 (LVR: 15.2 kb, RVR: 2.4 kb) after 44 passages on the MS cell line [27]; from strain ASFV_G (LVR: 7 kb, RVR: 5 kb), after 60 passages on Vero cells [28]; and from the Vero-adapted strain BA71V (LVR: 1.7 and 2.7 kb, RVR: 8 kb) [29]. In all cases, the deletions mainly affected genes of the MGF100, MGF360, and MGF505 families.

The Lv17/WB/Rie1 genome is not stable in Cos7 cells. After eight passages of Lv17/WB/Rie1 in Cos7 cells, an indel variant became the major component of the viral quasispecies. A deletion of approximately 40 kb was detected at the 5′ end of the genome, resulting in the loss of 52 genes. However, a similarly sized insertion from the 3′ end almost restored the genome to its original size and resulted in a de facto ~40 kb ITR at the ends of the genome, duplicating 51 genes. Such loss of the left distal 5′ end and its replacement with a 3′ end sequence is not without example; a similar but smaller duplicative translocation was observed in the genome of ASFV Estonia, in which a 7.3 kb duplication of the 3′ end replaced a 14.5 kb deletion at the 5′ end [30]. The puzzling questions of why such a huge translocation occurs and why it is beneficial for viral replication in Cos7 cells require further research and await answers.

As advantageous as this mutation may be in tissue culture, it took only three passages in PAM cells for the dominant variant of Cos7 cells (Lv17_cos8) to become undetectable and an entirely different genetic variant (Lv17_cos8_pam3) to dominate the quasispecies, which produced a seemingly homogeneous stock after three other passages (Lv17/WB/Rie1-Δ24) (Figure 1A). To our knowledge, a large contiguous deletion such as that carried by Lv17/WB/Rie1-Δ24 is rare in viruses replicating in PAM. One has been reported with a similar size (~24.5 kb of ASFV-∆LVR v. ~26.5 kb of Lv17/WB/Rie1-Δ24), and it also significantly reduced (~10-fold) the yield of the ASFV-∆LVR in PAM cells [31].

An analysis of the different mutants in their respective hosts revealed that the ORFs of Lv17/WB/Rie1 between regions of 2352 and 26,554 are not necessary for in vitro replication, either in PAM or in Cos7 cells. However, in their absence, the virus is unable to induce protective immunity in pigs even after repeated, relatively high-dose (10^2^ FFU followed by 10^4^ FFU) inoculations. It is unlikely that these MGF proteins are direct targets of the protective immune response [32,33,34]. Insufficient protection is more likely due to decreased in vivo replication due to the lack of proteins encoded by these genes. On the other hand, the region between 26,554 and 40,568 must contain (MGF 360-11L, MGF 360-12L, MGF 360-13L, MGF 360-14L, MGF 505-2R, MGF 505-3R, MGF505-4R, MGF 505-5R, MGF 505-6R) important genes for efficient in vitro replication in PAM, at least in the absence of genes encoded by fragments 2352–26,554. Interestingly, these genes appear to be non-essential or even detrimental to the replication of ASFV in Cos7.

The sequencing pipeline used by us and others [10,25,35] is obviously capable of detecting single-nucleotide changes or even polymorphisms if they are dominant in the quasispecies, but it is not suitable for the precise characterization of low-frequency variants, even at relatively high coverage. Due to the additive distorting effects of whole-genome amplification and the systematic biases and inconsistencies intrinsic to the NGS process [36,37,38,39,40,41], it is very difficult to estimate the ratio of the minor/major genetic components or even detect them in the different viral passages. *As a consequence of this methodical disproportionality, in general, it cannot be determined with absolute certainty whether a given virus population is genetically homogeneous or what the genetic makeup of the minor subpopulation is, even if high coverage of a viral sample is achieved.* Nothing illustrates this better than the fact that we could only detect a single read covering the uniquely distinctive region (26,555–40,568) of the Lv17_cos8_pam3 mutant in the eighth Cos7 passage, despite it becoming the major component in three passages in PAMs and a large number of reads (~2.8 million) being examined, resulting in a relatively high average coverage (2274) of the major component (Lv17_cos8) genome (Figure 1B). On the other hand, the detection of even a small number of viral reads that do not match the genome of the major component strongly indicates the presence of genetic variants in the quasispecies (if laboratory contamination is excluded).

Even though the sequences of some virus isolates [21,42,43,44] and the success of techniques used to produce recombinant viruses [20,35,45] clearly indicate that homologous recombination occurs between different ASFV strains, very little is known about this phenomenon in ASFV. To our knowledge, our work is the first to provide direct experimental evidence that homologous recombination of ASFV occurs in macrophages.

Homologous recombination is often triggered by a double-strand break. The repair of double-strand breakage has been extensively studied in both prokaryotes and eukaryotes, and, indeed, all experimental results and derived models indicate that the core of the process is strand invasion, in which a protein-associated single-stranded DNA from the broken end invades a similar or identical recipient double-stranded DNA [46]. However, single-stranded DNA is generated not only during DNA repair but also during DNA synthesis, and bacterial studies have shown that members of the homologous recombination protein machinery play multiple roles at the replication forks [47]. We can only speculate about the cause and mechanism of the relatively high recombination rate observed in PAM, but given the primary site of ASFV genome replication [48], it is very likely that viral proteins, rather than host proteins, play a major role in this process. Indeed, several viral proteins have been identified that are orthologs of proteins known to be involved in homologous recombination (e.g., single-stranded DNA-binding protein (pCP312R) [49], lambda-like exonuclease (pD345L) [50], topoisomerase II (pP1192R) [51], Nfo-like AP endonuclease (pE296R) [38] and other proteins of the base excision repair system (pC962R (FCS-like finger DNA primase)) [52], pO174L (DNA polymerase X) [53], NP419L (ligase) [54], etc.). But which of these are actually involved in recombination, and to what extent, can only be determined experimentally.

The ease with which a recombinant virus can be isolated from a crossing of two viruses carrying fluorescent markers separated by a distance of just over 10% of the genome suggests that, if coinfection occurs, homologous recombination occurs with relatively high frequency. The fact that at least two recombination events are clearly detectable in the genome of the recombinant virus by unique parental genetic markers (Figure 2) strongly supports the previous statement. A relatively simple method, namely the crossing of viruses carrying fluorescent markers that facilitate the detection and isolation of recombinants, will allow us to study homologous recombination between different ASFV strains in vitro and in vivo in the future. These experiments may not only answer theoretical questions but may also be of crucial importance for the regulation and safe use of live, attenuated virus vaccines (to limit the occurrence of in vivo recombination between vaccine strains or field and vaccine strains).

An effective vaccine against African swine fever has not yet been developed; the most promising approach seems to be the development of modified live virus-based (MLV) vaccines with targeted gene modifications. The rational design of an MLV is complicated by the fact that the effect of eliminating individual genes can vary significantly from one virus strain to another; the effect of multiple gene deletions is not linearly additive and is rather unpredictable. For example, the deletion of the EP153R gene from a naturally occurring, non-HAD, and non-lethal isolate (ASFV/NH/P68) results in a loss of protection against unrelated viruses [55]. The deletion of the EP402R gene leads to the attenuation of the BA71 strain and protection against the parental virus [56]. However, knocking out the EP402R and BL119 genes from a highly virulent Georgia/07 isolate resulted in an attenuated phenotype but lost protection compared to the version from which only the B119L gene was deleted [57].

pEP402R and pEP153R are not essential for virus replication in cells, and the results show that using a previously attenuated virus (by deleting the DP148R gene from the Benin isolate), EP402R and EP153R play a synergistic role in further reducing clinical symptoms and the virulence and persistence of the virus in the blood [58]. In contrast, the combined knockout of three genes (B119L, EP402R, and EP153R) from the ASFV-G isolate significantly reduced the protective potential compared to when only the B119L gene was removed from the ASFV-G isolate [59]. A similar loss of protection was associated with a reduction in in vitro (Figure 3) and in vivo replication capacity (Table 5), when the same three genes were deleted from the Lv17/WB/Rie1 genome.

Pigs inoculated with both vaccine candidate strains showed several mild clinical signs and elevated body temperature (Figure 4 and Figure 5), despite the fact that ASFV was undetectable until 4 DPC. An increase in clinical signs was detected after the booster vaccination, with a higher viral load in both strains, which increased after the challenge infection. Overall, fewer symptoms were observed in the Lv17/WB/Rie1-Δ24 group than in the Lv17/WB/Rie1-ΔCD-ΔGL group (Figure 5). Despite the viruses causing clinical symptoms, measurable antibody levels were only induced in one animal in the Lv17/WB/Rie1-Δ24 group by the third week after the first dose (Table 4). The second higher dose induced relatively low serological immunity in four animals, while in the Lv17/WB/Rie1-ΔCD-ΔGL group, only one animal proved to be responsive. Unfortunately, the unacceptably high death rate after the challenge confirms the obvious: that these viruses, in their current form, are not suitable for use as vaccines.

## 5. Conclusions

After eight passages of Lv17/WB/Rie1 in Cos7 cells, an indel variant (Lv17_cos8) became the main component of the viral quasispecies. The left ~40 kb terminal fragment of the genome was replaced with a similarly sized fragment translocated from the right side of the genome, resulting de facto in a ~40 kb ITR. After three low-dilution passages on PAM, Lv17_cos8 became undetectable, and a completely different genetic variant (Lv17_cos8_pam3) emerged as the major component of the viral quasispecies. Two more high-dilution passage and one low-dilution passage rendered Lv17_cos8_pam3 to be seemingly homogeneous, resulting in the Lv17/WB/Rie1-Δ24 virus containing a deletion between nucleotides 187 and 26554. The experiment illustrates that tissue culture-maintained ASFV strains can exist as quasispecies that rapidly shift toward a new master sequence and a new mutant spectrum upon replication in altered host cells.

Crossing two Lv17/WB/Rie1 mutants (Lv17/WB/Rie1-ΔCD and Lv17/WB/Rie1-ΔGL) by co-infection in PAM resulted in the isolation of recombinant Lv17/WB/Rie1-ΔCD-ΔGL. It demonstrates that recombination can and does occur between different ASFV strains in macrophages and that recombinant strains can be generated and isolated relatively easily by ex vivo virus crossing. The identification of unique parental markers on the Lv17/WB/Rie1-ΔCD-ΔGL genome indicates at least two recombination events during the crossing, which suggests that homologous recombination is a relatively frequent event in the ASFV genome during replication in PAM.

Pigs infected with the Lv17/WB/Rie1-Δ24 and Lv17/WB/Rie1-ΔCD-ΔGL strains have shown mild clinical signs, despite the fact that ASFV could not be detected in their blood until the challenge with the Armenia/07 strain.

Unfortunately, not even high-dose (10^4^ FFU) inoculations of these two viruses were able to induce protective immunity in the majority of the tested animals; therefore, they cannot be used as vaccines in their current form.

## Figures and Tables

**Figure 1 vaccines-11-01860-f001:**
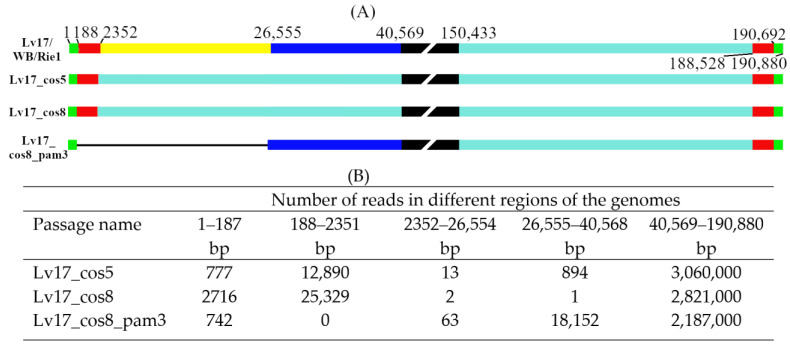
Graphical and numerical visualization of nucleotide coverage of passaged Lv17/WB/Rie1 genomes. (**A**) Four graphical panels represent the genome of the parental Lv17/WB/Rie1, Lv17_cos5, Lv17_cos8, and Lv17_cos8_pam3 genomes (respectively). The broken black boxes represent the central parts of the genomes (not proportional). Colored bars correspond to important structural genome regions determined by the following numbers: 188 and 26,555 deletion breakpoints in Lv17_cos8_pam3; 2352 and 188,528 proximal end of ITRs in the parental and Cos7 passaged viruses; 40,569 proximal end of the terminal deletion (2532–40,569) in the left variable region of Lv17_cos5, Lv17_cos8; 150,433 proximal end of the duplicated terminal fragment (150,433–190,880) in the right variable region of Lv17_cos5, Lv17_cos8. All numbering corresponds to the Lv17/WB/Rie1 sequence. (**B**) Number of reads in the five significant regions determined by the endpoints of ITRs and the deletions.

**Figure 2 vaccines-11-01860-f002:**
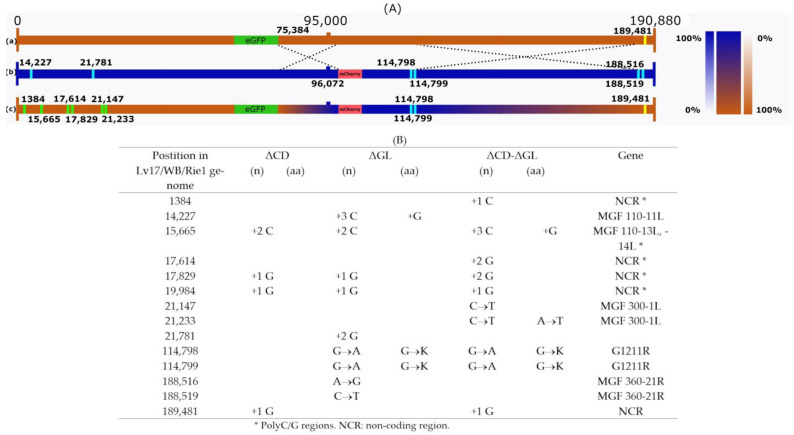
Schematic representation of the Lv17/WB/Rie1-ΔCD-ΔGL and its parental genomes. (**A**) The genomes of Lv17/WB/Rie1-ΔCD (a) and Lv17/WB/Rie1-ΔGL (b) are illustrated by horizontal brown and blue bars, respectively. Recombinant Lv17/WB/Rie1-ΔCD-ΔGL (c) is depicted by brown–blue mosaic bars. Color-transition regions (75222-96072 and 114799-189481) represent the areas where recombination (symbolized by dashed crossing lines) must have occurred. Vertical color-coded transition bars (right side) scale the probability of a nucleotide originating from one or the other parental genome within the two recombination regions. Numbered yellow and blue vertical bars represent distinguishing SNPs in the two parental genomes, while green vertical bars indicate unique mutations in Lv17/WB/Rie1-ΔCD-ΔGL. (**B**) Unique mutations (SNPs) in the three viral genomes. In all cases, numbering is according to the Lv17/WB/Rie1 genome.

**Figure 3 vaccines-11-01860-f003:**
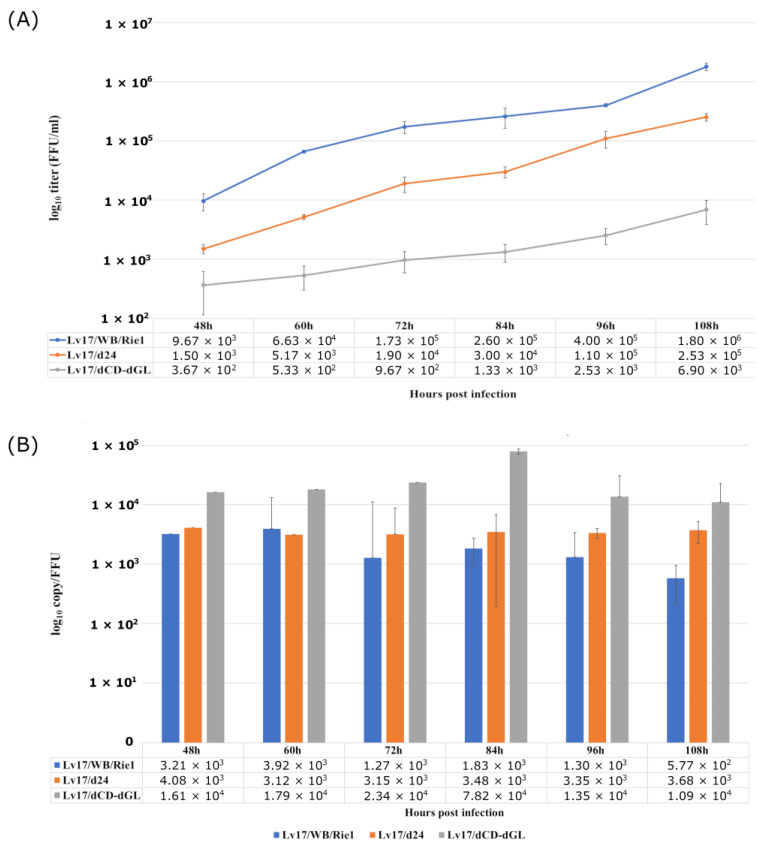
Growth characteristics of the mutant viruses Lv17/WB/Rie1-Δ24 and Lv17/WB/Rie1-ΔCD-ΔGL and their parental strain Lv17/WB/Rie1. PAMs were infected (MOI = 0.01), aliquots were taken every 12 h from 48 h to 108 h, and virus yields were determined by titration. (**A**) Mean values and standard errors of three parallel experiments. (**B**) Variations in specific infectivity (calculated as genome copy/mL to FFU/mL ratio).

**Figure 4 vaccines-11-01860-f004:**
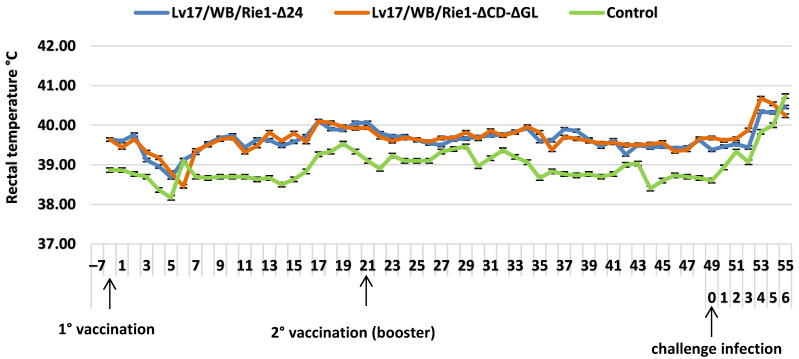
Rectal temperatures (mean values + standard errors) of animals injected with Lv17/WB/Rie1-Δ24 and Lv17/WB/Rie1-ΔCD-ΔGL of African swine fever virus (ASFV) and challenged with Armenia/07 of ASFV.

**Figure 5 vaccines-11-01860-f005:**
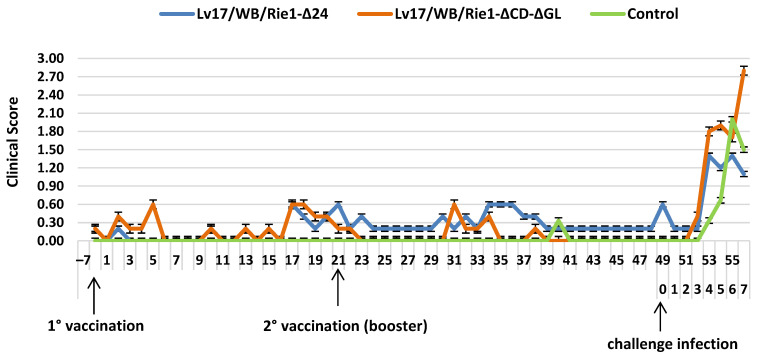
Clinical scores (mean values + standard errors) observed in different groups injected with Lv17/WB/Rie1-Δ24 and Lv17/WB/Rie1-ΔCD-ΔGL of African swine fever virus (ASFV) and challenged with Armenia/07 of ASFV.

**Table 1 vaccines-11-01860-t001:** African swine fever (ASF) vaccine candidate used in the experiment.

No. of Pigs	Vaccine Candidate	Virus Concentration in FFU * in One Dose (2 mL) of Vaccination		Inoculation Route
		First Administration	Second Administration	
5	Lv17/WB/Rie1-Δ24	10^2^	10^4^	Intramuscular **
5	Lv17/WB/Rie1-ΔCD-ΔGL	10^2^	10^4^	Intramuscular
3	*Unvaccinated controls*			

* FFU, fluorescent focus unit; ** into the neck muscle (right side).

**Table 2 vaccines-11-01860-t002:** Deleted and duplicated genes in Lv17_cos5, Lv17_cos8, and Lv17_cos5_pam3.

**1-40569.**
Deleted genes (52) from Lv17_cos5 and Lv17_cos8:
DP60L, KP93L, MGF_360-1L, MGF_360-2L, KP177R, L83L, L60L, MGF_360-3L, MGF_110-1L, MGF_110-2L, MGF_110-3L, MGF_110-4L, MGF_110-5L/6L, MGF_110-7L, MGF_110-8L, MGF_100-1R, MGF_110-9L, MGF_110-11L, MGF_110-12L, MGF_110-13L, MGF_110-14L, MGF_360-4L, MGF_360-6L, X69R, MGF_300-1L, MGF_300-4L, MGF_360-8L, MGF_360-9L, MGF_360-10L, MGF_360-11L, MGF_505-1R, MGF_360-12L, MGF_360-13L, MGF_360-14L, MGF_505-2R, MGF_505-3R, MGF_505-4R, MGF_505-5R, MGF_505-6R, ASFV_G_ACD_01990R, ASFV_G_ACD_00090, ASFV_G_ACD_00120, ASFV_G_ACD_00160, ASFV_G_ACD_00190, ASFV_G_ACD_00210, ASFV_G_ACD_00240, ASFV_G_ACD_00270, ASFV_G_ACD_00300, ASFV_G_ACD_00320, ASFV_G_ACD_00330, ASFV_G_ACD_00360, ASFV_G_ACD_00520
**150433-190880.**
Duplicated genes (51) in Lv17_cos5 and Lv17_cos8:
P1192R, H359L, H171R, H124R, H339R, H108R, H233R, H240R, R298L, Q706L, QP509L, QP383R, E184L, E183L, E423R, E301R, E146L, E199L, E165R, E248R, E120R, EP296R, E111R, E66L, I267L, I226R, I243L, I73R, I329L, I78L, I215L, I177L, I196L, DP238L, MGF_360-16R, DP63R, MGF_505-11L, MGF_100-1L, L7L, L8L, L9R, L10L, L11L, MGF_360-18R, DP71L, DP96R, MGF_360-19R, MGF 360-21R, DP93R, DP60R
**188-26555.**
Deleted genes (41) from Lv17_cos8_pam3:
DP60L, KP93L, MGF_360-1L, MGF_360-2L, KP177R, L83L, L60L, MGF_360-3L, MGF_110-1L, MGF_110-2L, MGF_110-3L, MGF_110-4L, MGF_110-5L/6L, MGF_110-7L, MGF_110-8L, MGF_100-1R, MGF_110-9L, MGF_110-11L, MGF_110-12L, MGF_110-13L, MGF_110-14L, MGF_360-4L, MGF_360-6L, X69R, MGF_300-1L, MGF_300-4L, MGF_360-8L, MGF_360-9L, MGF_360-10L, ASFV_G_ACD_01990R, ASFV_G_ACD_00090, ASFV_G_ACD_00120, ASFV_G_ACD_00160, ASFV_G_ACD_00190, ASFV_G_ACD_00210, ASFV_G_ACD_00240, ASFV_G_ACD_00270, ASFV_G_ACD_00300, ASFV_G_ACD_00320, ASFV_G_ACD_00330, ASFV_G_ACD_00360

**Table 3 vaccines-11-01860-t003:** Characterization of macroscopic lesions in different groups injected with Lv17/WB/Rie1-Δ24 and Lv17/WB/Rie1-ΔCD-ΔGL of African swine fever virus (ASFV) and challenged with Armenia/07 of ASFV.

Group	Pig	Lymph Nodes ^†^	Tonsils	Lungs	Spleen	Kidney	Heart	Skin ^††^	Intestine
Lv17/WB/Rie1-Δ24	1	+/a	-	-	+/b	+/e	+/e	+/g	+/h
	2	+/a	-	-	+/b	+/e	+/c, e	+/g	+/h
	3	+/a	-	-	+/b	+/f	+/f	+/g	-
	4	+/a	-	+/d	+/b	+/e	+/e	+/g	-
	5	+/a	+/e	+/d	+/b	-	+/c	-	-
Lv17/WB/Rie1-ΔCD-ΔGL	6	+/a	+/e	-	+/b	+/f	-	-	-
	7	+/a	-	+/d	+/b	-	-	+/g	+/h
	8	+/a	+/e	+/d	+/b	+/a	-	-	
	9	+/a	-	-	+/b	-	-	+/g	-
	10	+/a	+/e	-	+/b	-	+/e	+/g	+/h
*Control*	11	+/a	-	-	-	+/e	+/c	-	-
	12	+/a	-	+/d	+/b	+/e	+/e	-	-
	13	+/a	-	+/d	+/b	+/e	+/e	-	-

^†^ Bronchial, submandibular, epigastric, perirenal, mesenteric, superficial inguinal; ^††^ head, limbs, buttocks, ears; a, enlargement and hemorrhagic; b, splenomegaly; c, pericardial effusion; d, acute broncopneumonia; e, superficial point hemorrhages; f, petechial hemorrhages; g, diffuse hemorrhages; h, hemorrhagic enteritis of the small intestine.

**Table 4 vaccines-11-01860-t004:** Antibody response of pigs immunized with Lv17/WB/Rie1-Δ24 and Lv17/WB/Rie1-ΔCD-ΔGL of African swine fever virus (ASFV) and challenged with Armenia/07 of ASFV.

Strain/ Controls	Days Post-Vaccination (DPV)							Days Post-Challenge (DPC)		
	0	6	10	17	21	31	35	0 *	4 **	7 ***
Lv17/WB/Rie1-Δ24	-	-	-	^b^ 1 ^a^ (31.2%)	1 (23.5%)	4 (27.2%)	4 (26.2%)	4 (9.4%)	5 (1.5%)	5 (1.5%)
Lv17/WB/Rie1-ΔCD-ΔGL	-	-	-	-	-	1 (39.7%)	1 (39.5%)	1 (39.5%)	1 (17.4%)	4 (9.4%)
*Control*	-	-	-	-	-	-	-	-	-	-

* 49 DPV; ** 53 DPV; *** 56 DPV; ^a^ positive detection ratio obtained by commercial ELISA test (ID Screen^®^ African Swine Fever Competition, Grenoble, France); ^b^ number of pigs (average competition percentage value).

**Table 5 vaccines-11-01860-t005:** Detection of the ASFV genome by real-time PCR in pigs immunized with Lv17/WB/Rie1-Δ24 and Lv17/WB/Rie1-ΔCD-ΔGL viruses and challenged with ASFV Armenia/07.

Strain/ Controls	Days Post-Vaccination (DPV)							Days Post-Challenge (DPC)		
	0	6	10	17	21	31	35	0 *	4 **	7 ***
Lv17/WB/Rie1-Δ24	-	-	-	-	-	-	-	-	^b,a^ 1 (37.8)	5 (28.5)
Lv17/WB/Rie1-ΔCD-ΔGL	-	-	-	-	-	-	-	-	3 (35.9)	5 (27.1)
*Controls*	-	-	-	-	-	-	-	-	2 (27.1)	2 (18.8)

WOAH Manual of Diagnostic Tests and Vaccines for Terrestrial Animals; * 49 DPV; ** 53 DPV; *** 56 DPV; ^a^ positive detection ratio obtained by real-time PCR; ^b^ number of pigs (average of threshold cycles value).

## Data Availability

Data are contained within the article.

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
