# Peer review of "The Production of Recombinant African Swine Fever Virus Lv17/WB/Rie1 Strains and Their In Vitro and In Vivo Characterizations"

_vaccines, 2023, doi:10.3390/vaccines11121860_

Round 1

Reviewer 1 Report (Previous Reviewer 2)

Comments and Suggestions for Authors

This study explored the gene mutations of ASFV Lv17/WB/Rie1 during the cell culture passages, the recombination between ASFV strains in vitro, and the immune protection of two deletion mutant ASFV against virulent ASFV challenge. It was found that 1, in different cell cultures, the different ASFV quasispecies quickly adapt to various cells; 2, ASFV recombination occurs in co-infected PAMs; 3, two gene deleted ASFV Lv17/WB/Rie1 strains were significantly attenuated in vitro and in vivo, but was not able to protect against virulent Armenia/07 challenge. Quite a few work has been done and it gives an interesting information. Generally, it is a valuable work despite with a number of questions.

1, Lines 325-326, it mentioned that the stock was not homogeneous, was the parent ASFV Lv17/WB/Rie1 purified by plaque assay?

2, In the challenge experiment, the virulent Armenia/07 was used. How about the homology between ASFV Lv17/WB/Rie1 and Armenia/07? Is the parent strain of ASFV Lv17/WB/Rie1 immunization able to protect from virulent Armenia/07 challenge?

3, The sequencing information of Lv17/WB/Rie1-Δ24 as well as Lv17/WB/Rie1-ΔCD-ΔGL also should be mentioned in the Methods.

4, Line 203, the statistical analysis should be arranged at the end of Methods.

5, Line 307, should the (1-40568) be (2352-40568)?

6, Lines 388, 570, the ex vivo should be in vitro.

7, Lines 426-427, the sentence is really confusing, it should be restated.

8, Fig 4, here the arrow should be at 21 DPV.

9, Tables 3 and 4, the “a” and “b” should be correctly labelled.

Additionally, the description of results especially of the animal experiments should be better organized and more concise.

Author Response

1, Lines 325-326, it mentioned that the stock was not homogeneous, was the parent ASFV Lv17/WB/Rie1 purified by plaque assay?

The parental Lv17/WB/Rie1 was originally isolated on PBMC (https://pubmed.ncbi.nlm.nih.gov/30667598/), its sequence (GeneBank Accession Number: OR863253) was found “homogenic” without deletion variants.

2, In the challenge experiment, the virulent Armenia/07 was used. How about the homology between ASFV Lv17/WB/Rie1 and Armenia/07? Is the parent strain of ASFV Lv17/WB/Rie1 immunization able to protect from virulent Armenia/07 challenge?

Armenia/07 is 99.7% homologous to Lv17/WB/Rie1. We inserted this into the methods and material section as it follows: (Line 235) infection with the Armenia/07 strain of ASFV (99.7% homology to Lv17/WB/Rie1).

Yes ASFV Lv17/WB/Rie1 is able to protect against Armenia/07 and Georgia 2007 (https://www.mdpi.com/2076-393X/11/4/846  https://pubmed.ncbi.nlm.nih.gov/30667598/.)

3, The sequencing information of Lv17/WB/Rie1-Δ24 as well as Lv17/WB/Rie1-ΔCD-ΔGL also should be mentioned in the Methods.

We included the following modification in the text:

“(Line 198) Full genome sequencing of all strains and isolates was performed according to Olasz et al., (2019) [25].”

4, Line 203, the statistical analysis should be arranged at the end of Methods.

Thank you for your suggestion, we rearranged the M&M.

5, Line 307, should the (1-40568) be (2352-40568)?

Text was modified as it follows (Line 310-311): “the major viral component of the quasispecies lost around 40 kb (2352-40568) of its left terminus”

6, Lines 388, 570, the ex vivo should be in vitro.

We corrected them in the text (Lines 389 and 578).

7, Lines 426-427, the sentence is really confusing, it should be restated.

We restated the two sentences (Line 435-437): ”Four days after challenge, the Lv17/WB/Rie1-Δ24 and Lv17/WB/Rie1-ΔCD-ΔGL groups showed fever (> 40.5°C). While fever (Figure 4) was observed in the control group at 6 DPC.

8, Fig 4, here the arrow should be at 21 DPV.

Arrow position was corrected.

9, Tables 3 and 4, the “a” and “b” should be correctly labelled.

We corrected the labelling (Now in tables 4 and 5).

Additionally, the description of results especially of the animal experiments should be better organized and more concise.

We believe that the tables and figures will be in the right place after upmaking the final print.

Reviewer 2 Report (Previous Reviewer 1)

Comments and Suggestions for Authors

After the edits, the article became quite acceptable for publication

Author Response

Dear Reviewer,

Thank you for your suggestions!

Reviewer 3 Report (Previous Reviewer 3)

Comments and Suggestions for Authors

The authors attempted to address my comments, but unfortunately the genome sequence of the parental wild type virus is still missing. This again precludes the interpretation of the results.

I would also recommend the authors to include the data from supplementary table 1 in the manuscript itself. Instead of a table, or perhaps in addition to the table, a diagram depicting the modifications on the genome will facilitate viewing. For example, something similar to figure 4 in: Zani L, Forth JH, Forth L, Nurmoja I, Leidenberger S, Henke J, Carlson J, Breidenstein C, Viltrop A, Höper D, Sauter-Louis C, Beer M, Blome S. (2018). Deletion at the 5'-end of Estonian ASFV strains associated with an attenuated phenotype. Sci Rep. Apr 25;8(1):6510 or figure 6 in: Wang, T., Wang, L., Han, Y., Pan, L., Yang, J., Sun, M., Zhou, P., Sun, Y., Bi, Y., & Qiu, H.-J. (2021). Adaptation of African swine fever virus to HEK293T cells. Transboundary and Emerging Diseases, 68, 2853–2866.

Comments on the Quality of English Language

N/A

Author Response

The authors attempted to address my comments, but unfortunately the genome sequence of the parental wild type virus is still missing. This again precludes the interpretation of the results.

In the last days we have been informed by the lab sequencing LV17 that the sequence is now available and we have included it in the text as it follows: (Line 125) „Cos7 cells were infected with ASFV Lv17/WB/Rie1 (GeneBank Accession Number: OR863253)

I would also recommend the authors to include the data from supplementary table 1 in the manuscript itself. Instead of a table, or perhaps in addition to the table, a diagram depicting the modifications on the genome will facilitate viewing. For example, something similar to figure 4 in: Zani L, Forth JH, Forth L, Nurmoja I, Leidenberger S, Henke J, Carlson J, Breidenstein C, Viltrop A, Höper D, Sauter-Louis C, Beer M, Blome S. (2018). Deletion at the 5'-end of Estonian ASFV strains associated with an attenuated phenotype. Sci Rep. Apr 25;8(1):6510 or figure 6 in: Wang, T., Wang, L., Han, Y., Pan, L., Yang, J., Sun, M., Zhou, P., Sun, Y., Bi, Y., & Qiu, H.-J. (2021). Adaptation of African swine fever virus to HEK293T cells. Transboundary and Emerging Diseases, 68, 2853–2866.

We included a new table (now Table 2) in the manuscript.

Round 2

Reviewer 3 Report (Previous Reviewer 3)

Comments and Suggestions for Authors

N/A

Comments on the Quality of English Language

N/A

This manuscript is a resubmission of an earlier submission. The following is a list of the peer review reports and author responses from that submission.

Round 1

Reviewer 1 Report

Comments and Suggestions for Authors

Article entitled “The production of recombinant ASFV Lv17/WB/Rie1 strains and their characterizations as vaccine candidates” has some scientific value.

1.      The main problem of the article is the lack of final result since the study of the virus strains turned out to be unable to induce protective immunity in relation to the virulent strain. As a result, the article has an unfinished appearance. With this in mind, it may be more correct to describe less the possibilities of vaccination and focus more on the data obtained on the recombination of viral strains

2.      In my opinion, the most interesting data were obtained when identifying a recombinant variant of the virus (Lv17/WB/Rie1-ΔCD-ΔGL.). Unfortunately, the authors do not sufficiently consider the reasons for the appearance of this variant of the virus and the mechanism of its occurrence. Nevertheless, it would be highly desirable to describe possible mechanisms, at least speculatively.

3.      Figure 3 should contain quantitative data on viral levels not only of new strains of the virus, but also comparison with a known isolate (for example Armenia07).

4.      As investigated ASFV strains in their current form are not suitable for use as vaccines the title of the article should be changed.

5.      There are a lot of unnecessary details (for example, injections in the right side of the neck) about vaccination attempts.

6.      It is necessary to provide a standard for the ratio of the Fluorescence Focus Unit with copies of genomes detected by the rtPCR method, to allow a clearer understanding of the doses of viruses.

7.      It would be desirable to explain the reason for using Cos7 cells for virus isolation followed by replication on PAM. Why weren't PAMs used to isolate the virus?

Reviewer 2 Report

Comments and Suggestions for Authors

In this manuscript, two kinds of recombinant ASFV Lv17/WB/Rie1 strains were produced. One is Lv17/WB/Rie1-Δ24 produced by illegitimate recombination mediated by low dilution serial passage in Cos7 cell line and isolated on PAM cell culture. Another one is ASFV Lv17/WB/Rie1-ΔCD-ΔGL, generated by homologous recombination crossing two ASFV strains (Lv17/WB/Rie1-ΔCD and Lv17/WB/Rie1-ΔGL containing eGFP and mCherry markers) during PAM co-infection. The above work revealed that ASFV was easily subjected to deletion and insertion in continuous cell culture, different ASFV quasispecies fit in various cell cultures, and ASFV recombination easily occurs in co-infected PAMs by different ASFV strains. Further work showed that these two types mutant ASFVs replicate less efficiently in PAMs, and was not able to provide protection against virulent ASFV challenge in pigs.

In general, the work is pretty informative and couple of interesting phenomenon were revealed.  

There are following concerns and suggestion for further improvement:

1, The paragraph of lines 379-393. Here it says the mutant viruses have decreased SI, but later on the results showed the higher SI values of mutant viruses. The two aspects are conflicted with each other and need to be reconciled and corrected.

2, The temperature in each groups in Fig 4 and the clinical scores in each groups of Fig 5 are all average values? If so, where are the error bars?

3, Lines 425-426, “During the entire vaccination period, the clinical score of Lv17/WB/Rie1-Δ24 and Lv17/WB/Rie1-ΔCD-ΔGL groups remained below 0.6”. The statement is not true.

4, Other minors

A, lines 37-38, on the market mortality despite that it can reach 100%, should be “on the market despite that it can reach 100% mortality,”

B, line 216, candidate should be candidates.

C, line 281, “proximal end of the terminal deletion (1-40569) in the left variable region of Lv17_cos5, Lv17_cos8;” should be “proximal end of the terminal deletion (2532-40569) in the left variable region of Lv17_cos5, Lv17_cos8;”

D, Lines 296 and 298, the right terminus, left terminus should be left terminus, right terminus.

E, line 453, what is the f ?

Reviewer 3 Report

Comments and Suggestions for Authors

The manuscript by Petrini et al., describes the generation of two different ASFV deletion mutants using the Lv17/WB/Rie1 strain as parental virus.  This is a naturally attenuated isolate with the potential to be used as a live attenuated vaccine against ASF and therefore the results presented here are of high interest to the field. Unfortunately, crucial data is missing and this limits interpretation of the results. Specifically, throughout the manuscript genome locations of major deletions are given but these are pointless in the absence of a reference genome. I strongly recommend the authors to follow the Journal guidelines regarding deposition of sequences: “New nucleic acid sequences must be deposited into an acceptable repository such as GenBank, EMBL, or DDBJ. Sequences should be submitted to only one database”. This would allow the identification of the deleted genes from the virus genome (a table with this information is also required) and further our understanding on how ASFV adapts to grow in continuous cell lines such as Cos7 cells.  

Other major points:

Lines 77-78:  This sentence is controversial since recent data has shown that EP153R is not required for HAD (Perez-Nunez et al., 2023)

Lines 442-443: It is not clear when the animals were culled. Did they reach the pre-established humane end point?

Comments on the Quality of English Language

N/A
